# The Regulatory Role of Apelin on the Appetite and Growth of Common Carp (*Cyprinus Carpio* L.)

**DOI:** 10.3390/ani10112163

**Published:** 2020-11-20

**Authors:** Xiao Yan, Chaobin Qin, Guokun Yang, Dapeng Deng, Liping Yang, Junchang Feng, Jiali Mi, Guoxing Nie

**Affiliations:** College of Fisheries, Henan Normal University, Xinxiang 453007, China; yanxiaobangong@163.com (X.Y.); qinchao88639501@163.com (C.Q.); 2018026@htu.edu.cn (G.Y.); 2013093@htu.edu.cn (D.D.); weedyung@163.com (L.Y.); 2016026@htu.edu.cn (J.F.); mjl211@163.com (J.M.)

**Keywords:** apelin, appetite, growth, *Cyprinus carpio* L.

## Abstract

**Simple Summary:**

Food intake and growth are two interrelated physiological processes. Fish obtain energy from the outside to meet their own growth and development through food intake. Apelin, as a new endocrine factor, plays a wide range of biological roles. Among them, apelin’s role in regulating appetite and growth has attracted the attention of researchers. Based on our investigations, in vitro and in vivo experiments showed that Pyr-apelin-13 could induce significant changes in mRNA levels of appetite-related and growth-related genes, suggesting that Pyr-apelin-13 may regulate the appetite and growth of common carp by regulating the expression of these key genes.

**Abstract:**

Apelin, a kind of active polypeptide, has many biological functions, such as promoting food intake, enhancing immunity, and regulating energy balance. In mammals, studies have indicated that apelin is involved in regulating food intake. However, there are relatively few studies about the regulatory effect of apelin on fish feeding, and the specific mechanism is not clear. Therefore, the purpose of this study was to preliminarily investigate the regulatory effects of apelin on key genes of feeding and growth in common carp (*Cyprinus Carpio* L.) through in vitro and in vivo experiments. In the present study, after incubation with different concentrations of Pyr-apelin-13 (0, 10, 100, and 1000 nM) in hypothalamic fragments, the expressions of *Neuropeptide Y* (*NPY*) and *Agouti related peptide* (*AgRP*) mRNA were significantly up-regulated at 12 and 3 h, respectively, and the significant down-regulation of *Cocaine and amphetamine-related transcript* (*CART*) mRNA expression was observed at 1 and 3 h. In vivo, after Pyr-apelin-13 oral administration (0, 1, 10, and 100 pmol/g), the *orexin* mRNA level in the hypothalamus of common carp was significantly increased at 1, 6, and 12 h, while *CART/*(*Proopiomelanocortin*) *POMC* mRNA levels in the hypothalamus of common carp were significantly down-regulated. Following incubation with different concentrations of Pyr-apelin-13 (0, 10, 100, and 1000 nM) in primary hepatocytes, *GHR* (*Growth hormone receptor*), *IGF2* (*Insulin-like growth factor 2*), *IGFBP2* (*Insulin like growth factor binding protein 2*), and *IGFBP3* (*Insulin like growth factor binding protein 3*) mRNA levels were significantly increased at 3 h. In vivo, the levels of *IGF1* (*Insulin-like growth factor 1*), *IGF2*, *IGFBP2* (*Insulin like growth factor binding protein 2*), and *IGFBP3* mRNA were significantly increased after the oral administration of Pyr-apelin-13 in the hepatopancreas, in a time and dose-dependent manner. These results support the hypothesis that Pyr-apelin-13 might regulate the feeding and growth of common carp through mediating the expressions of appetite- and growth-related genes. Overall, apelin, which is an orexigenic peptide, improves food intake and is involved in the growth of common carp.

## 1. Introduction

Apelin is an endocrine regulator, and its receptor is APJ (a putative receptor protein related to the angiotensin receptor AT1) [1]. Apelin cDNA encodes pre-proapelin, which is composed of 77 amino acids, with a signal peptide consisting of 22 amino acids at its *N*-terminal. After removing signal peptide and enzyme digestion, apelin exists in many active forms, mainly including apelin-36, apelin-17, apelin-13, and the pyroglutamate apelin-13 (Pyr-apelin-13). Pyr-apelin-13, apelin-13, and apelin-36 have comparable efficacy and potency in human cardiovascular tissues and chronic metabolic disease caused by the disturbance of carbohydrate metabolism, whereas apelin-17 appears to be the most efficient in promoting apelin receptor internalization [2,3]. In terms of amino acid composition, the 12 consecutive amino acids of goldfish (*Carassius auratus*) [4], common carp (*Cyprinus Carpio* L.) [5], and Ya-fish (*Schizothorax prenanti*) [6] at the C-terminal are exactly the same as those of vertebrates, which are also in the apelin-13 region. Thus, this conserved sequence of 13 amino acids is an important site for apelin to bind to its receptor and display its biological activity. In addition, compared with other apelin peptides, Pyr-apelin-13 has stronger affinity with the APJ receptors [7], and its *N*-terminal pyroglutamate may prevent the degradation of exopeptidase [8]. Therefore, in in vitro and in vivo experiments, Pyr-apelin-13 or apelin-13 was mostly used to study the physiological functions of apelin. In mammals and teleosts, apelin and its receptor APJ are widely expressed in central nervous system and peripheral tissues [4,9,10]. In addition, it has been found that apelin is highly expressed in the brain, especially in the hypothalamus. The hypothalamus is an important part of the brain, which can secrete a variety of appetite-regulating factors, suggesting that apelin may play an important role in the regulation of fish feeding [11,12].

Studies in mammals have shown that apelin is involved in the regulation of food intake. For example, intracerebroventricular (*i.c.v.*) injection of Pyr-apelin-13 (30 nM) significantly increased the food intake at 24 h after rats fasted, which showed the same result as that in Sprague Dawley rats [13,14]. Compared with mammals, there are relatively few studies on the effect of apelin on food intake in fish. In goldfish and red-bellied piranha (*Pygocentrus nattereri*), the expression of apelin mRNA was significantly up-regulated in the brain after fasting [15]. After intraperitoneal (*i.p.*) injection of apelin-13, the food intake of cavefish (*Astyanax fasciatus mexicanus*) [16], goldfish [4], Ya-fish [6], and Siberian sturgeon (*Acipenser baerii*) [10] increased significantly. These results show that apelin can promote the food intake of fish. In our other study (under review), common carp was fed with dietary supplementation of Pyr-apelin-13 for 8 weeks. The results showed that Pyr-apelin-13 could promote the food intake and growth of common carp because of the increase in weight gain rate (WGR), specific growth rate (SGR), and relative food intake (RFI).

Food intake and growth are two interrelated physiological processes. Animals obtain energy from the outside to meet their own growth and development through feeding [17]. The food intake and growth of fish are affected by endogenous and exogenous factors. The endogenous factors are mainly endocrine factors (appetite-regulating factors and growth-regulating factors). Orexin, AgRP, NPY, CART, and POMC are important appetite regulators secreted from the brain [18,19,20]. Their functions have been studied in fish. For example, after intraperitoneal injection of 100 ng Orexin for 30 min, food intake increased significantly in cavefish, suggesting that Orexin is an appetite stimulating factor [16]. It was found that the food intake of tilapia (*Oreochromis mossambicus*) was significantly increased after intraperitoneal injection of NPY (0.6 g/g) for 10 h [21]. AgRP is a neuropeptide that is expressed mainly in the hypothalamus and acts as an appetite stimulator. For example, AgRP gene expression in the hypothalamus of goldfish was significantly increased after 3, 5, and 7 days of starvation [22]. Both CART/POMC are appetite inhibitors, which have been confirmed in goldfish and rainbow trout (*Oncorhynchus mykiss*) [23,24]. Studies in fish have shown that apelin can regulate the expression of food intake-related factors. Therefore, the results suggested that apelin regulated fish food intake by regulating related endocrine factors [25,26]. Common carp (*Cyprinus carpio*) is a kind of omnivorous fish, which has the characteristics of fresh meat, rich nutrition, high yield, and economic value. It is an important freshwater breed in China. Apelin is an endocrine regulator, which has the biological function of promoting fish feeding and enhancing immunity, and has potential application value in aquaculture [27]. The regulation effect of apelin on growth and feeding heretofore has not been reported in common carp. Therefore, this paper takes common carp as the research object. Through in vitro incubation and gavage, the effects of Pyr-apelin-13 on food intake and growth were explored through the measurement of gene expression related to appetite and growth.

## 2. Materials and Methods

### 2.1. Experimental Animals

All experimental common carp (*n* = 600) were obtained from a local fishery (Xinxiang, China). Fish were acclimated for 2 weeks in an indoor tank (5 × 4 × 2 m^3^) and fed commercial pellets (Tong wei, China) three times daily until satiety before starting the trial [5]. Table 1 indicates the formulation and proximate composition of the diet.

This study was approved by the Institutional Animal Care and Use Committee of Henan Normal University and complied with the guidance of ethical animal treatment for the care and use of experimental animals. To reduce the pain of the experimental fish, we anesthetized them with 100 mg/L MS-222 (Sigma, St. Louis, MO, USA) before each treatment.

### 2.2. Reagents

Pyr-apelin-13 (Pyr-PRPRLSHKGPMPF) was commercially synthesized by Sangon Biotech (Shanghai, China). High-performance liquid chromatography (HPLC) was used to analyze and identify the purity of the polypeptide, which was confirmed to be >98% [5].

### 2.3. Gavage Treatment of Pyr-Apelin-13

Ninety-six common carp with body weight of 97.55 ± 8.32 g were used for gavage administration of the Pyr-apelin-13 test, and they were randomly divided evenly among 16 tanks. After the end of acclimation, the experimental fish that fasted 24 h were anesthetized with 100 mg/L MS-222 (Sigma, St. Louis, MO, USA), and orally administrated Pyr-apelin-13 (0, 1, 10, and 100 pmol/g, body weight). After Pyr-apelin-13 treatment for 1, 3, 6, and 12 h, fish (6 replicates per group) were euthanized using an overdose of MS-222 [28] and decapitated. Hypothalamus and hepatopancreas were sampled quickly and frozen by liquid nitrogen and stored at −80 °C until analysis.

### 2.4. Hypothalamus Preparation and Treatments

Experimental fish (*n* = 96, 29.18 ± 2.31 g) were sacrificed with a severed head. The hypothalamus was quickly separated and placed into precooled Ca^2+^/Mg^2+^-free HBSS. On the ultra-clean workbench, the hypothalamus was rinsed with Ca^2+^/Mg^2+^-free HBSS three times to remove impurities, such as blood clots. The hypothalamus was divided into two pieces and then preincubated with DMEM/F-12 (Gibco, Gaithersburg, MD, USA) medium containing 10% FBS (Hyclone, Logan, UT, USA) for 6 h at 28 °C and no CO_2_ [29]. After preincubation, the old medium was removed and the fragments of hypothalamus were rinsed twice with DMEM/F12 (without FBS), and then 500 μL DMEM/F-12 medium (Gibco, Gaithersburg, MD, USA) was added. After 1 h, 500 μL medium (containing Pyr-apelin-13) was added into each well, with final concentrations of Pyr-apelin-13 (0, 10, 100, and 1000 nM), respectively. After incubation for 1, 3, 6, and 12 h, the hypothalamus fragments were analyzed by an RNAiso Plus (TaKaRa, Shiga, Japan), and quickly stored at −80 °C until analysis.

### 2.5. Isolation, Culture and Treatments of Primary Hepatocytes

Fish (29.18 ± 2.31 g) were anesthetized with 100 mg/L MS-222 (Sigma, St. Louis, MO, USA), and the hypothalamus was divided and placed in ice-cold culture medium (Ca^2+^/Mg^2+^-free HBSS). The hepatocytes were separated and cultured as described previously [5]. The hepatic cells, density of 3 × 10^5^ cells/mL, were cultured overnight at 28 °C and no CO_2_. After overnight, the medium was replaced with serum-free fresh medium, and the hepatic cells were preincubated for 1 h. Finally, the hepatic cells were treated with various Pyr-apelin-13 (0, 10, 100, and 1000 nM) or PBS for 3 and 6 h, and analyzed by an RNAiso Plus (TaKaRa, Shiga, Japan), collected, and immediately stored at −80 °C until use.

### 2.6. RNA Extraction, Reverse-Transcription, and Real-Time Quantitative PCR

RNA extraction, cDNA synthesis, and real-time quantitative PCR were performed according to standard procedures established in our lab [3]. Total RNA of the hypothalamus and hepatopancreas was extracted by the RNAiso Plus Reagent (Takara, Shiga, Japan) according to the manufacturer’s instructions [5]. The concentration and purity of RNA were measured by Nanodrop 2000 (Thermo, Waltham, MA, USA). Then, 1 μg total RNA from hepatopancreas was treated with DNase I (RNase-free) and the first strand cDNA was synthesized with PrimeScript RT Enzyme (TaKaRa, Shiga, Japan). As for real-time quantitative PCR, specific primers were designed using Primer 5 software (Table 2). The PCR amplification was carried out using the LightCycler 480 II (Roche, Basel, Switzerland) with the SYBR^®^
*Premix Ex Tap*^TM^ II (Takara, Shiga, Japan). All reactions were performed in triplicate. The procedure was as follows: 10 s at 95 °C for initial denaturation, followed by 40 cycles of 95 °C for 5 s, 60 °C for 20 s. The relative mRNA level of the gene was analyzed by the 2^−^^△△CT^ method with 18S and β-actin as the internal control [5].

### 2.7. Statistical Analysis

The results were expressed as mean ± SEM (*n* = 6). All data were analyzed using the software SPSS 20.0 (IBM; Armonk, NY, USA), and the normality and homogeneity of data were checked and confirmed. Statistical differences between treatment and control groups at each time point were identified by one-way analysis of variance (ANOVA) followed by Duncan’s multiple range test. Significant difference was approved at *p* < 0.05 [5].

## 3. Results

### 3.1. Effects of Pyr-Apelin-13 Incubation on the Expression of Appetite-Related Genes in the Hypothalamic Debris

The effects of apelin on the expression of appetite-related genes of common carp hypothalamic debris were assessed (Figure 1). As for *orexin* mRNA expression, it was not influenced by Pyr-apelin-13 treatment (Figure 1A). The significant increase (*p* < 0.05) in *NPY* mRNA expression at 12 h was found by Pyr-apelin-13 with a concentration of 10 nM (Figure 1B). Pyr-apelin-13 with a concentration of 1000 nM significantly up-regulated (*p* < 0.01) *AgRP* mRNA expression after 3 h treatment (Figure 1C). However, the level of *CART* mRNA was markedly inhibited (*p* < 0.05) at 1 h post Pyr-apelin-13 (10, 1000 nM) administration, and at 3 h after Pyr-apelin-13 with the concentration of 10 nM (Figure 1D). As for *POMC* mRNA expression, it was not influenced by Pyr-apelin-13 treatment (Figure 1E).

### 3.2. Effects of Perfusing Pyr-Apelin-13 on the Expression of Appetite-Related Genes in the Hypothalamus

As shown in Figure 2, after common carp were given various doses of Pyr-apelin-13 (0, 10, 100, and 1000 pmol/g BW) by gavage, the expression of *orexin* mRNA in the hypothalamus was significantly (*p* < 0.05) up-regulated at 1, 6, and 12 h (Figure 2A). However, the levels of *NPY* and *AgRP* mRNA in the hypothalamus were not affected after Pyr-apelin-13 administration (Figure 2B, C). The level of *CART* mRNA was markedly significantly (*p* < 0.05) suppressed by 1 pmol/g Pyr-apelin-13 at 12 h (Figure 2D). The level of *POMC* mRNA was markedly inhibited (*p* < 0.05) at 1 h post Pyr-apelin-13 (10, 100 pmol/g) administration, and at 12 h after Pyr-apelin-13 with the concentrations of 1 and 10 pmol/g (Figure 2E).

### 3.3. Effects of Pyr-Apelin-13 Incubation on the Expression of Growth-Related Genes in the Primary Hepatocytes

In primary cultured hepatocytes, the effects of Pyr-apelin-13 on the mRNA expression of appetite-related genes were also assessed. As shown in Figure 3, 1000 nM Pyr-apelin-13 significantly (*p* < 0.05) stimulated the expression of *GHR* mRNA at 3 h (Figure 3A). There was no significant difference in the level of *IGF1* mRNA at 3 and 6 h after Pyr-apelin-13 treatment (Figure 3B). After primary hepatocytes were incubated with concentrations of 10 and 1000 nM Pyr-apelin-13, the level of *IGF2* mRNA was significantly (*p* < 0.01) up-regulated at 3 h, while it was down-regulated by 1000 nM Pyr-apelin-13 at 6 h (Figure 3C). The 1000 nM and 10 nM concentrations of Pyr-apelin-13 significantly (*p* < 0.05) stimulated the expression of *IGFBP2* (Figure 3D) and *IGFBP3* (Figure 3E) mRNA at 3 h, separately.

### 3.4. Effects of Perfusing Pyr-Apelin-13 on the Expression of Growth-Related Genes in the Hepatopancreas

In the hepatopancreas, there was no significant difference in *GHR* mRNA abundance among the different groups after being treated with Pyr-apelin-13 (Figure 4A). The level of *IGF1* mRNA in the hepatopancreas was significantly (*p* < 0.05) up-regulated at 1 h following 10 and 100 pmol/g Pyr-apelin-13 administration, and it was also significantly (*p* < 0.05) increased at 12 h by 1 pmol/g Pyr-apelin-13 (Figure 4B). The *IGF2* mRNA expression was significantly (*p* < 0.05) increased at 3 and 12 h in the hepatopancreas after 1 pmol/g Pyr-apelin-13 treatment (Figure 4C). The 10 and 1 pmol/g Pyr-apelin-13 significantly (*p* < 0.01) stimulated the expression of *IGFBP2* (Figure 4D) and *IGFBP3* (Figure 4E) mRNA at 6 and 12 h, separately.

## 4. Discussion

In this study, through in vitro and in vivo experiments, the regulation of Pyr-apelin-13 on key appetite-related genes (Orexin, NPY, AgRP, POMC, and CART) was investigated in common carp. Apelin and these peptides are widely distributed in the central nervous system, and it has been proved that there is a morphological connection between the systems in which these peptides are located. The morphological connection is the basis of their functional interaction [25].

Some studies have shown that orexin is an appetite-promoted polypeptide; apelin can regulate the mRNA expression of *orexin* and then, regulate fish feeding. For instance, in goldfish, Volkoff et al. [25] studied the effect of apelin on *orexin* mRNA expression in the hypothalamus, forebrain, and afterbrain in vitro. It was found that the expression of *orexin* mRNA increased significantly after incubation with 100 nM apelin-13 for 1 h. In vivo, studies have shown that Pyr-apelin-13 can increase food intake by up-regulating *orexin* mRNA expression of the hypothalamus [26]. In the *i.p.* injection of cavefish, apelin significantly increased the *orexin* mRNA level of the hypothalamus [19]. These results indicated that apelin can regulate fish feeding by up-regulating *orexin* mRNA expression. In our study, *orexin* mRNA expression of the hypothalamus increased significantly after Pyr-apelin-13 administration dose-dependently in vivo. However, Pyr-apelin-13 in this study has no influence on the level of *orexin* mRNA in vitro. The difference in results may be related to species-specific reasons, apelin concentration, apelin isoform (e.g., apelin-12, apelin-13, apelin-36, or Pyr-apelin-13), or treatment method.

It has been reported that NPY/AgRP play the appetite-promoting role in the food intake regulation of fish. In goldfish, apelin (100 nM) treatment induced an increase in NPY in goldfish, which is consistent with the results in our research [25]. In this study, the *NPY* mRNA level was significantly up-regulated in the hypothalamic debris incubated with 10 nM Pyr-apelin-13. However, in vitro, there is no significant change in NPY release of the rat hypothalamus incubated with 10 nM apelin-13 for 30 min [30]. The differences of these results may be related to species specificity. In this study, the expression of *AgRP* mRNA is significantly elevated in the hypothalamus after high concentration of Pyr-apelin-13 (1000 nM) administration for 3 h. These results suggested that apelin may affect food intake of common carp by promoting the mRNA expression of *AgRP* and *NPY*.

On the other hand, POMC/CART are appetite-suppressed polypeptides [15]. In the arcuate nucleus of the hypothalamus of rodents, apelin and its receptor are expressed in POMC/CART neurons, suggesting that apelin may affect the expression and secretion of CART [15]. In this study, the expression of *CART* mRNA was significantly inhibited at 1 and 3 h after Pyr-apelin-13 treatment, indicating that Pyr-apelin-13 can regulate the food intake of common carp by decreasing the expression of *CART* mRNA, which is in line with the results in mice [26]. In vivo, after given Pyr-apelin-13 by gavage, the mRNA expression level of *CART* in common carp was significantly down-regulated in a time- and dose-dependent manner. The above results showed that Pyr-apelin-13 could regulate the feeding of common carp through mediating the expression of *orexin*, *POMC/CART* and *NPY/AgRP* mRNA.

In fish, the study of apelin mainly focuses on cardiovascular development and feeding regulation, while the effects of apelin on growth has not been reported [10,22]. However, food intake and growth are inextricably linked. The growth of animals is regulated by the hypothalamus–pituitary–liver axis, and the hypothalamus secretes GHRH, which acts on the pituitary to promote the secretion of GH. Then, the combination of GH and GH receptor promotes the synthesis and secretion of IGFs in the liver, and regulates the growth of somatic cells [31,32]. To study the effect of Pyr-apelin-13 on the expression of growth-related genes in common carp, the experiment of Pyr-apelin-13 incubation of primary hepatocytes was carried out. In this study, the significant increase in *GHR* and *IGF2* mRNA levels was observed in hepatocytes after Pyr-apelin-13 treatment. In mice, the level of leptin in plasma increased significantly after apelin-13 treatment [13]. In tilapia, the expression of *IGF1*, *IGF2*, and *GHR* mRNA was significantly promoted in hepatocytes incubated with leptin for 18 h [33]. Therefore, it is a conjecture that the increase in *GHR* and *IGF2* mRNA expression may be caused by the elevation of leptin induced by apelin. In this study, we found that Pyr-apelin-13 induced the significant increase in *IGF1* and *IGF2* mRNA expression in the hepatopancreas, which may be related to the elevation of plasma leptin level caused by apelin administration [13,33]. Insulin-like growth factor binding proteins (IGFBPs) are regulatory proteins that bind to insulin-like growth factor (IGFs), regulating the ability of IGFs to bind to their receptors (IGFR). They affect the signal intensity in the downstream signal transduction pathway of IGFR and regulate the growth and proliferation of target cells [34]. Atlantic salmon (*Salmo salar*) parr were implanted intraperitoneally with cortisol implants (0, 10, and 40 μg/g body weight) and sampled after 3 or 14 days [34]. The results showed that hepatic *IGFBP1* and *IGFBP2* were stimulated by the high cortisol dose and indicated that cortisol modulates the growth of juvenile salmon via the regulation of hepatic *IGFBPs* [35]. In this study, Pyr-apelin-13 stimulated the expression of *IGFBP2* and *IGFBP3* both in vivo and in vitro. Therefore, combined with relevant research reports, it is speculated that Pyr-apelin-13 may regulate the growth of carp by regulating the expression of *IGFBPs*.

## 5. Conclusions

In conclusion, our research mainly focused on the regulation of Pyr-apelin-13 on the food intake and growth of common carp. Our results revealed that Pyr-apelin-13 supplementation promoted food intake and growth in common carp by regulating the mRNA expression levels of key genes. The research results deepen our understanding of the physiological function of apelin in fish, and provide theoretical support for further exploring the regulatory mechanism of fish feeding and growth. However, in the present study, we studied the regulation of Pyr-apeln-13 on feeding and growth-related genes of common carp through short-term oral administration and cell incubation in vitro. This is a tentative exploration. In order to further elucidate the regulation mechanism of Pyr-apelin-13 on feeding and growth, we fed the common carp with dietary supplementation of Pyr-apelin-13 for 8 weeks, in our other study. Then, we correlated the data of weight gain rate (WGR), specific growth rate (SGR), and relative food intake (RFI) with gene expression (not yet published). Furthermore, our future research direction is to study the signaling pathway of apelin regulation on feeding and growth of common carp at both the gene and protein levels.

## Figures and Tables

**Figure 1 animals-10-02163-f001:**
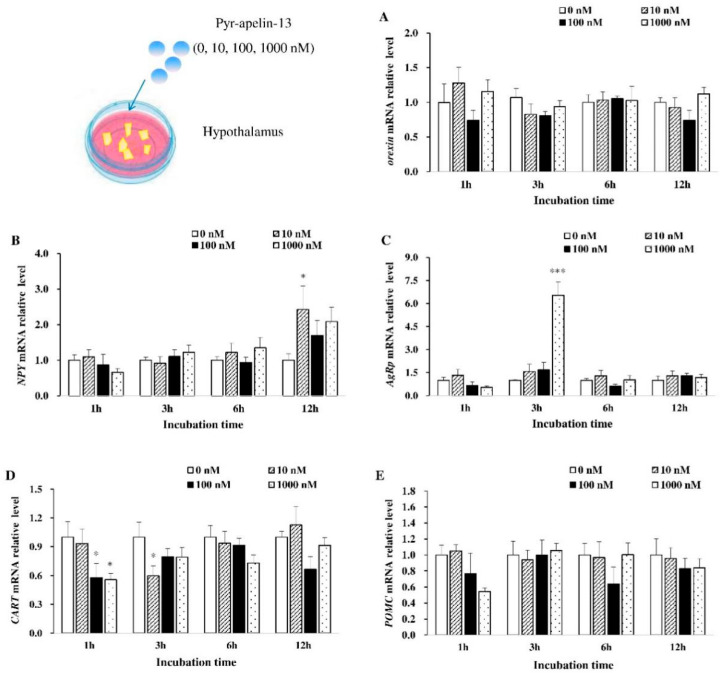
Effects of Pyr-apelin-13 (0, 10, 100, and 1000 nM) on mRNA expressions of appetite-related genes in *Cyprinus carpio* L. hypothalamic fragments. Orexin (**A**), NPY (**B**), AgRP (**C**), CART (**D**), POMC (**E**). Differences between treatment and control groups at each time point were tested with Duncan’s multiple range test, and are indicated by different asterisks (* *p* < 0.05, *** *p* < 0.001). There was no significant difference for PBS administration. The data were given as mean ± SEM. (*n* = 6).

**Figure 2 animals-10-02163-f002:**
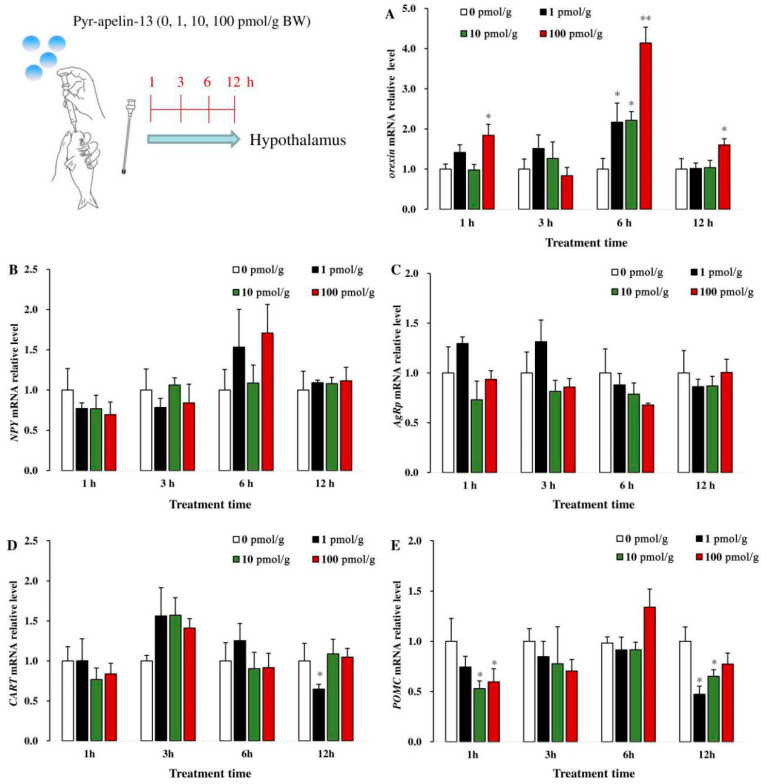
Gene expressions of appetite-related factors in the hypothalamus of *Cyprinus carpio* L. after gavage of Pyr-apelin-13 (0, 10, 100, and 1000 pmol/g). Orexin (**A**), NPY (**B**), AgRP (**C**), CART (**D**), POMC (**E**). Differences between treatment and control groups at each time point were tested with Duncan’s multiple range test, and are indicated by different asterisks (* *p* < 0.05, ** *p* < 0.01). The data were given as mean ± SEM. (*n* = 6).

**Figure 3 animals-10-02163-f003:**
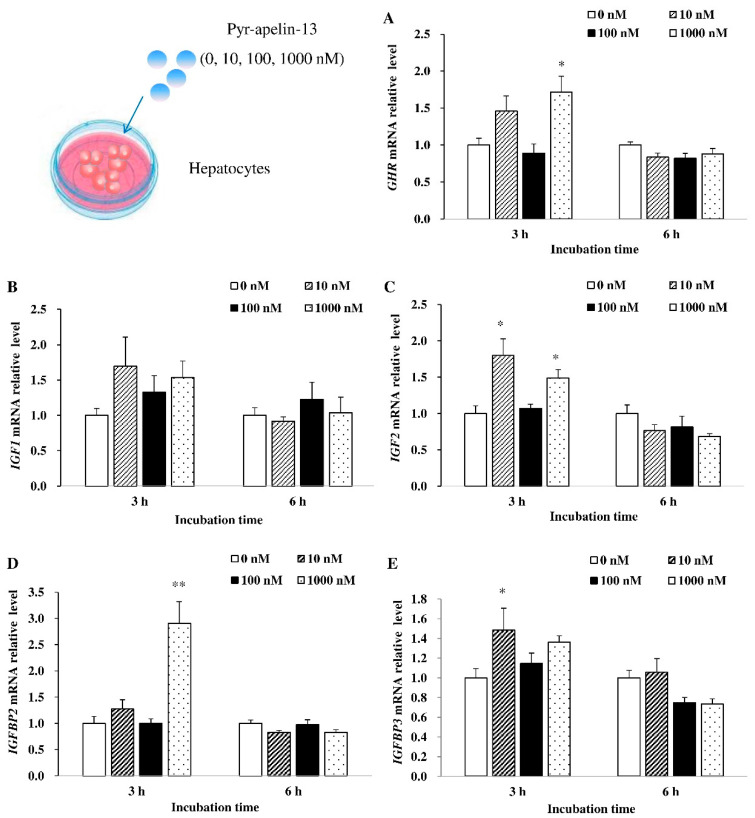
Effects of Pyr-apelin-13 (0, 10, 100, and 1000 nM) on mRNA expressions of growth-related genes in *Cyprinus carpio* L. primary hepatocytes. GHR (**A**), IGF1 (**B**), IGF2 (**C**), IGFBP2 (**D**), IGFBP3 (**E**). Differences between treatment and control groups at each time point were tested with Duncan’s multiple range test, and are indicated by different asterisks (* *p* < 0.05, ** *p* < 0.01). There was no significant difference for PBS administration. The data were given as mean ± SEM. (*n* = 6).

**Figure 4 animals-10-02163-f004:**
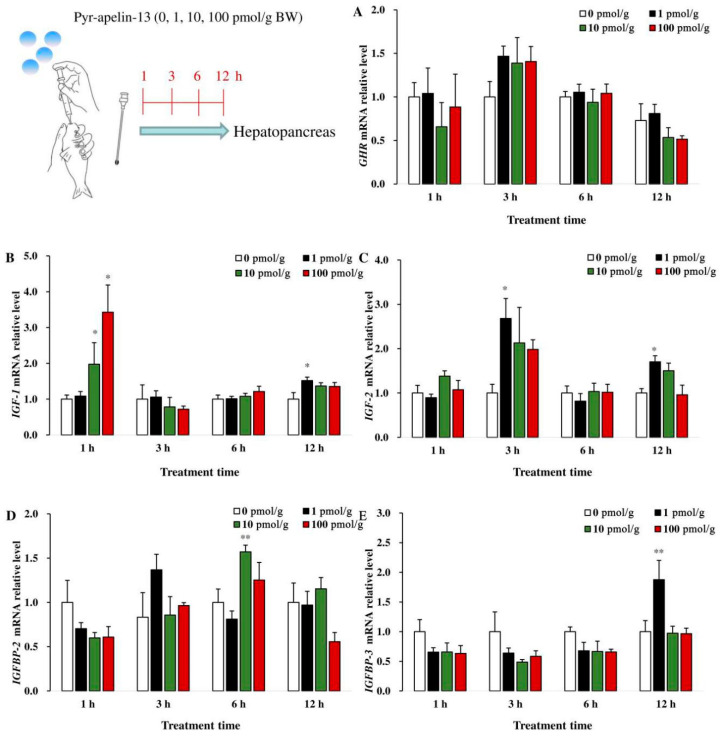
Gene expressions of growth-related factors in the hepatopancreas of *Cyprinus carpio* L. after gavage of Pyr-apelin-13 (0, 10, 100, and 1000 pmol/g). GHR (**A**), IGF1 (**B**), IGF2 (**C**), IGFBP2 (**D**), IGFBP3 (**E**). Differences between treatment and control groups at each time point were tested with Duncan’s multiple range test, and are indicated by different asterisks (* *p* < 0.05, ** *p* < 0.01). The data were given as mean ± SEM. (*n* = 6).

**Table 1 animals-10-02163-t001:** Formulation and proximate composition of the diet.

Ingredients	Contents (%)	Proximate Composition	Contents (%)
Fish meal	2.0	Moisture	9.05
Soybean meal	25.0	Crude lipid	9.56
Rapeseed meal	20.0	Crude protein	34.93
Rice bran meal	10.0	Ash	9.00
Rice bran	10.0	Nitrogen-free extract	29.26
Fish oil	2.0	Note: ^1^ Vitamin premix supplied the following vitamins (kg): V_A_ 800,000 IU, V_B1_ 1500 mg, V_B2_ 1250 mg, V_C_ 2.5 g, V_D3_ 160,000 IU, V_E_ 15 g, V_B12_ 4 mg, V_K3_ 325 mg, V_B6_ 1100 mg, Creatine 5.5 g, Folic acid 70 mg, Biotin 125 mg, Niacin 4 g, Pantothenic acid 4.5 g. ^2^ Mineral premix supplied the following minerals (kg): P 105 g, Ca 330 g, Mg 45 g, Fe 15 g, I 50 mg, Se 9 mg, Cu 0.35 g, Zn 3 g, Mn 1.5 g, Co 11 mg.	
Soybean oil	3.0		
Sesame meal	5.0		
Flour	10.0		
Wheat bran	10.0		
Ca(H_2_PO_4_)_2_	2.0		
Vitamin premix ^1^	0.2		
Mineral premix ^2^	0.6		
Choline chloride	0.2		
Total	100.0		

**Table 2 animals-10-02163-t002:** Primers used in qRT-PCR analyses.

Gene	Accession No.	Forward (5′→3′)	Reverse (5′→3′)	Product Length (bp)
*18S*	FJ710826.1	GAGACTCCGGCTTGCTAAAT	CAGACCTGTTATTGCTCCATCT	107
*β-actin*	M24113.1	TGCAAAGCCGGATTCGCTGG	AGTTGGTGACAATACCGTGC	293
*orexin*	XM_019086797.1	CGTCAAGGTCCTGCAAATTATAC	CGATAGCCGCGTCGTTATTA	105
*NPY*	XM_019063564.1	GCACTAAGACACTACATCAACCT	TGGGACTCTGTTTCACCAATC	104
*AgRP*	FR726954.1	GCACCACAACTCTGCATTAAC	GGTCTCACCACATGATGTCTC	101
*CART*	AM498379.1	TTCAGGGTGCCGAAATGG	CTGCTTCTCGTTGGTCAGATT	101
*POMC*	XM_019064968.1	GCTTCTACCACGCAGACTTTA	AAGGGCACATAGGTGCTAATC	101
*GHR*	AY741100.1	GAGTGATTGGAGTGGTGATTCT	GGTGCAGGAATAGGTGGTAAA	91
*IGF1*	KP661168.1	GGATATGGGCCTAGTTCAAGAC	TACGGGTGCACAATACATCTC	105
*IGF2*	AF402958.1	CCAGTTTCTATTCTTGCGGTTTC	CTGCGGCTTCTTTGTTCTTTC	105
*IGFBP2*	FJ009001.1	CCCACCTCTCCAATGATAAGG	GGGAAGGGATAGGAAGGTTTAG	113
*IGFBP3*	FJ424519.1	GCTGACCTCCCACTTGTATG	CACTCTCTCTGTCTCCTCTTCT	103

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
