# Peer review of "The Regulatory Role of Apelin on the Appetite and Growth of Common Carp (Cyprinus Carpio L.)"

_animals, 2020, doi:10.3390/ani10112163_

Round 1
Reviewer 1 Report
One typo needs to be corrected in line 53 in the Latin name of common carp
Author Response
Dear reviewer,
We appreciate the your positive and constructive comments on our manuscript (animals-1006220) entitled “The Regulatory Role of Apelin on the Appetite and Growth of Common Carp (Cyprinus Carpio L.)” The comments are valuable and helpful for revising and improving our paper, as well as the important guiding significance to our researches. According to your suggestions, we have made correction which we hope to meet with approval. The main corrections in the manuscript and the responses to your comments are as following.
Yours sincerely,
Dr. Guoxing Nie (Corresponding author)
(On behalf of all co-authors)
One typo needs to be corrected in line 53 in the Latin name of common carp.
Response:
Thanks for the reviewer’s kind advice. The typo has been corrected in line 55 in the Latin name of common carp. ‘In terms of amino acid composition, the 12 consecutive amino acids of goldfish (Carassius auratus) [4], common carp (Cyprinus Carpio L.) [5], and Ya-fish (Schizothorax prenanti) [6] at the C-terminal are exactly the same as that of vertebrates, which is also in the apelin-13 region.’
Reviewer 2 Report
Dear authors,
you have implemented several of the comments and recommendations, however, some are still in need of correction.
You includde some information about the statistics, however, just not into the statistics paragraph (2.7)...please change that and heed the comments more closely. Similarly, please describe unmistakenly, how you killed which fish. In section 2.3 it is still unclear, how the fish were killed. You mention, they were "anesthetized and smapled for hypothalamus and hepatopancreas" which in itself will kill the fish. If you did not specifically killed the fish, than indicate so.
Also, throughout the manuscript you are using "appetite-regulated" and "appetite-regulating". There is a huge difference between those two statements. The first one ("regulated") means that those genes are regulated by the appetite, while the latter one ("regulating" means, the genes are regulating the appetite. Please correct your manuscript carefully for these statements!
Figure 2 and 4 still need to be included in a better quality (it appears to me, that the quality has already been approved but not really a lot)!
In section 3.3 you are writing in the heading still about "appetite-related genes" but are providing data for growth-related genes...please let me not comment on this a third time!
The discussion has been approved and can stay the way it is.
Good luck with the corrections

Author Response
Dear reviewer,
We appreciate your positive and constructive comments on our manuscript (animals-1006220) entitled “The Regulatory Role of Apelin on the Appetite and Growth of Common Carp (Cyprinus Carpio L.)” The comments are valuable and helpful for revising and improving our paper, as well as the important guiding significance to our research. We are really sorry that the first modification did not fully meet your requirements. Therefore, according to your suggestion, we have carefully revised the manuscript a second time, and we hope to meet with approval. Based on your comments, we also attached a point-by-point letter to you. The main corrections in the manuscript and the responses to your comments are as following.
Yours sincerely,
Dr. Guoxing Nie (Corresponding author)
(On behalf of all co-authors)
Comments and Suggestions for Authors
Dear authors,
you have implemented several of the comments and recommendations, however, some are still in need of correction.
Response:
Thank you very much for your comments and recommendations on this manuscript. This has been a great help to our manuscript and research. In this revised manuscript, we revised it again according to your comments and recommendations, and hoping to meet your expectations.
You include some information about the statistics, however, just not into the statistics paragraph (2.7)...please change that and heed the comments more closely.
Response:
Thanks for the reviewer’s kind advice. We are very sorry for the inconvenience. We may not fully understand your comments when we first revise the manuscript. In this revised manuscript, We included more information about the statistics into the statistics paragraph (2.7). And we hope it will meet the requirements. ‘The results were expressed as Mean ± SEM (n = 6). All data were analyzed using the software SPSS 20.0 (IBM; Armonk, NY, USA), and the normality and homogeneity of data were checked and confirmed. Statistical differences between treatment and control groups at each time point were identified by one-way analysis of variance (ANOVA) followed by Duncan’s multiple rang test. Significant difference was approved at P < 0.05 [5].’
Similarly, please describe unmistakenly, how you killed which fish. In section 2.3 it is still unclear, how the fish were killed. You mention, they were "anesthetized and smapled for hypothalamus and hepatopancreas" which in itself will kill the fish. If you did not specifically killed the fish, than indicate so.
Response:
Thanks for the reviewer’s kind advice. Sorry again for the inconvenience. We may not fully understand your comments when we first revise the manuscript. In this revised manuscript, we added more details of experimental design. As shown below, ‘After Pyr-apelin-13 treatment for 1, 3, 6 h and 12 h, fish (6 replicates per group) were euthanized using an overdose of MS-222 [28] and decapitated. Hypothalamus and hepatopancreas were sampled quickly and frozen by liquid nitrogen and stored at −80 °C until analysis.’
Also, throughout the manuscript you are using "appetite-regulated" and "appetite-regulating". There is a huge difference between those two statements. The first one ("regulated") means that those genes are regulated by the appetite, while the latter one ("regulating" means, the genes are regulating the appetite. Please correct your manuscript carefully for these statements!
Response:
Thank you for pointing out our mistake. We are very sorry for such spelling mistakes. Therefore, we carefully examined the entire manuscript and made corrections where there were errors. The details are as follows,
Line 62-65, ‘And it has been found that apelin is highly expressed in the brain, especially in the hypothalamus. The hypothalamus is an important part of the brain, which can secrete a variety of appetite-regulating factors, suggesting that apelin may play an important role in the regulation of fish feeding [11, 12].’
Line 81-82, ‘The endogenous factors are mainly endocrine factors (appetite-regulating factors and growth-regulating factors).’
In section 3.3 you are writing in the heading still about "appetite-related genes" but are providing data for growth-related genes...please let me not comment on this a third time!
Response:
Thanks for the reviewer’s kind advice. We apologize for the spelling mistakes caused by our oversight. And we are also deeply sorry for any inconvenience caused to you. We have made corresponding changes according to your suggestion and changed ‘appetite-related genes’ to ‘growth-related genes’ in section 3.3.
Figure 2 and 4 still need to be included in a better quality (it appears to me, that the quality has already been approved but not really a lot)!
Response:
Thanks for the reviewer’s kind advice. We are very sorry for any inconvenience to you due to the quality of the pictures in the manuscript. The size of the image is compressed when it is put into WORD, which is the reason why the image is not so clear. When we submitted the manuscript, we also submitted clear pictures in PNG format according to the requirements of the journal. The sizes of Figure 2 and Figure 4 are 8815 pixels×8836 pixels (1.67 MB) and 8000 pixels×8522 pixels (1.61 MB), respectively. We have provided you with Figure 2 and Figure 4 in PDF format. Please refer to the attachment.
The discussion has been approved and can stay the way it is.
Good luck with the corrections
Response:
Thank you again for your help with this manuscript. Your detailed suggestions help us avoid mistakes and make our research and manuscript more scientific. We sincerely hope that the revised manuscript meets the requirements of reviewer and journal.

This manuscript is a resubmission of an earlier submission. The following is a list of the peer review reports and author responses from that submission.
Round 1
Reviewer 1 Report
This is my review of Manuscript ID: animals-952292. I recommend accept the following the article after correcting all my comments.
Summary and Abstract
Lines
10 - add - Apelin is a peptide?
16-33. Any abbreviation for the name given for the first time in the article should accurately write the entire name and incubate the abbreviation. E.g. Neuropeptide Y (NPY)?
30-33. Please write these lines carefully. The results support the hypothesis that ....
Introduction
37 More explanation is needed on the Apelin structure and function of the research topic.
42 More explanation is needed on the other apelin peptides structure and function.
57 Please explain the different between Pyr-apelin to apelin.
67-76. Please have an accurate explanation of the difference in research between fish and mammals rather than general words.
79-132. I suggest adding a reference to every chapters described in the holes material and methods: Experimental animals, Reagents, Gavage treatment of Pyr-apelin-13 Hypothalamus preparation and treatments, Statistical analysis.
Results
All the figures has to be described in details in the text.
Fig.1 and Fig 2 3 and 4. I please add explanation (ANOVA) followed by Tukey’s multiple rang test? (n = 6, *P < 0.05, **P < 0.01, ***P < 0.001). It is not exactly clear the significant differences between the treatment levels and the control (ANOVA) or Tukey’s between each level? Please explain.
Discussion
The discussion is an important part of the article but is in no way a repetition of the findings in other words or information about studies done and featured in the introduction. This discussion should be corrected so that you are clear about the meaning of the findings, the difference between them and the previous knowledge or they support it, the contribution of the work and suggestions for improvement.
193-198. These sentences are general information that has a place in the introduction and not in the discussion. Please forward this information if it did not appear for discussion.
199-201 These sentences are a repetition of what has already been written..
214-223 . Please whenever information of findings from other studies is presented in a discussion in these lines or elsewhere in the discussion they should be attributed to what is found in this work. Are they different from what is found in this study, or do they support the findings.
234-244. Important information but no clear connections to this work.
259-264. I suggested to add to the conclusion that described the results quite good also the weaknesses of the work and future studies need to improve the knowledge.
Author Response
Dear reviewer,
We appreciate the positive and constructive comments made by reviewers and editors on our manuscript (animals-952292) entitled “The Regulatory Role of Apelin on the Appetite and Growth of Common Carp (Cyprinus Carpio L.)” The comments are valuable and helpful for revising and improving our paper, as well as the important guiding significance to our researches. According to your suggestions, we have made correction which we hope to meet with approval. Based on your comments, we also attached a point-by-point letter to you. The main corrections in the manuscript and the responses to your comments are as following.
Yours sincerely,
Dr. Guoxing Nie (Corresponding author)
(On behalf of all co-authors)
Summary and Abstract
Lines
10 - add - Apelin is a peptide?
Response:
Thanks for the reviewer’s question. Apelin is a peptide and exists in many active forms, mainly including apelin-36, apelin-17, apelin-13 and the pyroglutamate apelin-13 (Pyr-apelin-13).
16-33. Any abbreviation for the name given for the first time in the article should accurately write the entire name and incubate the abbreviation. E.g. Neuropeptide Y (NPY)?
Response:
Thanks for the reviewer’s kind advice. In the revised manuscript, we have accurately writed the entire name and incubated the abbreviation for the name given for the first time in the article. Neuropeptide Y (NPY), Agouti related peptide (AgRP), (Proopiomelanocortin) POMC, Cocaine and amphetamine-related transcript (CART), (Growth hormone receptor) GHR, (Insulin-like growth factor 1) IGF1, (Insulin-like growth factor 2) IGF2, (Insulin like growth factor binding protein 2) IGFBP2, (Insulin like growth factor binding protein 3) IGFBP3, putative receptor protein related to the angiotensin receptor AT1 (APJ).
30-33. Please write these lines carefully. The results support the hypothesis that ....
Response:
Thanks for the reviewer’s kind advice. In the revised manuscript, we revised the sentence as follows ‘These results support the hypothesis that Pyr-apelin-13 might regulate the feeding and the growth of common carp through mediating the expressions of appetite- and growth-related genes.’
Introduction
37 More explanation is needed on the Apelin structure and function of the research topic.
Response:
Thanks for the reviewer’s kind advice. We have made corresponding changes according to your suggestion and rewrote this section. ‘Apelin is an endocrine regulator, and its receptor is APJ (putative receptor protein related to the angiotensin receptor AT1) [1]. Apelin cDNA encodes preproapelin, which is composed of 77 amino acids, with a signal peptide consisting of 22 amino acids at its N-terminal. After removing signal peptide and enzyme digestion, apelin exists in many active forms, mainly including apelin-36, apelin-17, apelin-13 and the pyroglutamate apelin-13 (Pyr-apelin-13).’ ‘In terms of amino acid composition, the 12 consecutive amino acids of goldfish (Carassius auratus) [4], common carp (Cyprimnus Carpio L.) [5], and Ya-fish (Schizothorax prenanti) [6] at the C-terminal are exactly the same as that of vertebrates, which is also in the apelin-13 region. Thus, this conserved sequence of 13 amino acids is an important site for apelin to bind to its receptor and display its biological activity.’
42 More explanation is needed on the other apelin peptides structure and function.
Response:
Thanks for the reviewer’s kind advice. We have made corresponding changes according to your suggestion and rewrote this section. ‘Pyr-apelin-13, apelin-13 and apelin-36 have comparable efficacy and potency in human cardiovascular tissues and chronic metabolic disease caused by disturbance of carbohydrate metabolism, whereas apelin-13 appears to be the most efficient in promoting apelin receptor internalization [2, 3]. In terms of amino acid composition, the 12 consecutive amino acids of In terms of amino acid composition, the 12 consecutive amino acids of goldfish (Carassius auratus) [4], common carp (Cyprimnus Carpio L.) [5], and Ya-fish (Schizothorax prenanti) [6] at the C-terminal are exactly the same as that of vertebrates, which is also in the apelin-13 region. Thus, this conserved sequence of 13 amino acids is an important site for apelin to bind to its receptor and display its biological activity.’
57 Please explain the different between Pyr-apelin to apelin.
Response:
Thanks for the reviewer’s question. As described above, the preproapelin (77 amino acids) is cleaved into proapelin (55 amino acids) by endopeptidases in rich basic amino-acids sites. The proapelin is then cleaved into various biologically active forms of apelin such as apelin-36, apelin-17 and apelin-13. Apelin-13 can be posttranslationally modified by the transformation of glutamine (Q) in the N-terminal position into pyroglutamine, forming the pyroglutamated apelin-13 (Pyr(1)-apelin-13), and the N-terminal pyroglutamate may prevent the degradation of apelin-13 by exopeptidase (as shown in the figure below).
Fig. 1 Apelin amino acid sequence and maturation (Castan-Laurell et al., 2012)
Castan-Laurell I, Dray C, Knauf C, et al. Apelin, a promising target for type 2 diabetes treatment?. Trends in Endocrinology & Metabolism, 2012, 23(5): 234-241.
67-76. Please have an accurate explanation of the difference in research between fish and mammals rather than general words.
Response:
Thanks for the reviewer’s kind advice. We have made corresponding changes according to your suggestion and rewrote this section. In the revised manuscript, we focus on the use of these proteins in fish and provide detailed examples. ‘The feed intake and growth of fish are affected by endogenous and exogenous factors. The endogenous factors are mainly endocrine factors (appetite-regulated factors and growth-regulated factors). Orexin, AgRP, NPY, CART and POMC are important appetite regulators secreted from brain [18-20]. Their functions have been studied in fish. For example, after intraperitoneal injection of 100 ng Orexin for 30 min, feed intake increased significantly in cavefish, suggesting that Orexin is an appetite stimulating factor [16]. It was found that the feed intake of tilapia (Oreochromis mossambicus) was significantly increased after intraperitoneal injection of NPY (0.6 g/g) for 10 h [21]. AgRP is a neuropeptide that is expressed mainly in the hypothalamus and acts as an appetite stimulators. For example, AgRP gene expression in the hypothalamus of goldfish was significantly increased after 3, 5, and 7 days of starvation [22]. Both CART/POMC are appetite inhibitors, which have been confirmed in goldfish and rainbow trout (Oncorhynchus mykiss) [23, 24].’
79-132. I suggest adding a reference to every chapters described in the holes material and methods: Experimental animals, Reagents, Gavage treatment of Pyr-apelin-13 Hypothalamus preparation and treatments, Statistical analysis.
Response:
Thanks for the reviewer’s kind advice. We have made corresponding changes according to your suggestion and added a reference to every chapters described in the holes material and methods: Experimental animals, Reagents, Gavage treatment of Pyr-apelin-13 Hypothalamus preparation and treatments, Statistical analysis.
Results
All the figures has to be described in details in the text.
Fig.1 and Fig 2 3 and 4. I please add explanation (ANOVA) followed by Tukey’s multiple rang test? (n = 6, *P < 0.05, **P < 0.01, ***P < 0.001). It is not exactly clear the significant differences between the treatment levels and the control (ANOVA) or Tukey’s between each level? Please explain.
Response:
Thanks for the reviewer’s kind advice. We have made corresponding changes according to your suggestion and rewroted this section.
‘Fig. 1. Effects of Pyr-apelin-13 (0 nM, 10 nM, 100 nM, 1000 nM) on mRNA expressions of appetite related genes in Cyprinus carpio L hypothalamic fragments. Differences between treatment and control groups at each time point were tested with Duncan’s multiple range test, and are indicated by different asterisks (*P < 0.05, **P < 0.01, ***P < 0.001). There was no significant difference for PBS administration. The data were given as Mean ± SEM. (n = 6)’
‘Fig. 2. Gene expressions of appetite related factors in the hypothalamus of Cyprinus carpio L after gavage of Pyr-apelin-13 (0 pmol/g, 10 pmol/g, 100 pmol/g, 1000 pmol/g). Differences between treatment and control groups at each time point were tested with Duncan’s multiple range test, and are indicated by different asterisks (*P < 0.05, **P < 0.01, ***P < 0.001). The data were given as Mean ± SEM. (n = 6)’
‘Fig. 3. Effects of Pyr-apelin-13 (0 nM, 10 nM, 100 nM, 1000 nM) on mRNA expressions of growth related genes in Cyprinus carpio L primary hepatocytes. Differences between treatment and control groups at each time point were tested with Duncan’s multiple range test, and are indicated by different asterisks (*P < 0.05, **P < 0.01, ***P < 0.001). There was no significant difference for PBS administration. The data were given as Mean ± SEM. (n = 6)’
‘Fig. 4. Gene expressions of growth related factors in the hepatopancreas of Cyprinus carpio L after gavage of Pyr-apelin-13 (0 pmol/g, 10 pmol/g, 100 pmol/g, 1000 pmol/g). Differences between treatment and control groups at each time point were tested with Duncan’s multiple range test, and are indicated by different asterisks (*P < 0.05, **P < 0.01, ***P < 0.001). The data were given as Mean ± SEM. (n = 6)’
Discussion
The discussion is an important part of the article but is in no way a repetition of the findings in other words or information about studies done and featured in the introduction. This discussion should be corrected so that you are clear about the meaning of the findings, the difference between them and the previous knowledge or they support it, the contribution of the work and suggestions for improvement.
193-198. These sentences are general information that has a place in the introduction and not in the discussion. Please forward this information if it did not appear for discussion.
Response:
Thanks for the reviewer’s kind advice. We have made corresponding changes according to your suggestion and deleted this section in the revised manuscript.
199-201 These sentences are a repetition of what has already been written.
Response:
Thanks for the reviewer’s kind advice. We deleted the duplicate and rewrote this section. ‘Apelin and these peptides are widely distributed in the central nervous system, and it has been proved that there is a morphological connection between the systems in which these peptides are located. The morphological connection is the basis of their functional interaction [25].’
214-223 . Please whenever information of findings from other studies is presented in a discussion in these lines or elsewhere in the discussion they should be attributed to what is found in this work. Are they different from what is found in this study, or do they support the findings.
Response:
Thanks for the reviewer’s kind advice. We have made corresponding changes according to your suggestion and rewrote this section. ‘It has been reported that NPY/AgRP play the appetite-promoted role in feed intake regulation of fish. In goldfish, the significant elevation of NPY mRNA level in the brain was found post the fasting, and NPY injection treatment notably promoted the increase of the feed intake .apelin (100 nM) treatment induced an increase in NPY in goldfish, which is consistent with the results in our researh In this study, NPY mRNA level was significantly up-regulated in the hypothalamic debris incubated with 10 nM Pyr-apelin-13, which is consistent with the results in goldfish [25]. In this study, NPY mRNA level was significantly up-regulated in the hypothalamic debris incubated with 10 nM Pyr-apelin-13. However, in vitro, there is no significant change in NPY release of the rat hypothalamus incubated with 10 nM apelin-13 for 30 min [29]. The differences of these results may be related to species specificity. In this study, the expression of AgRP mRNA is significantly elevated in the hypothalamus after high concentration of Pyr-apelin-13 (1000 nM) administration for 3 h. These results suggested that apelin may affect feed intake of common carp by promoting the mRNA expression of AgRP and NPY.’
234-244. Important information but no clear connections to this work.
Response:
Thanks for the reviewer’s kind advice. We have made corresponding changes according to your suggestion and deleted the content irrelevant to this article.
259-264. I suggested to add to the conclusion that described the results quite good also the weaknesses of the work and future studies need to improve the knowledge.
Response:
Thanks for the reviewer’s kind advice. We have made corresponding changes according to your suggestion and rewrote this section. ‘In conclusion, our research mainly focused on the regulation of Pyr-apelin-13 on the feed intake and growth of common carp. Our results revealed that Pyr-apelin-13 supplementation promoted feed intake and growth in common carp by regulating the mRNA expression levels of key genes. The research results deepen our understanding of the physiological function of apelin in fish, and provide theoretical support for further exploring the regulatory mechanism of fish feeding and growth. However, in the present study, we studied the regulation of Pyr-apeln-13 on feeding and growth related genes of common carp through short-term oral administration and cell incubation in vitro. This is a tentative exploration. In order to further elucidate the regulation mechanism of Pyr-apelin-13 on feeding and growth, we fed the common carp with dietary supplementation of Pyr-apelin-13 for 8 weeks, in our another study. Then we correlation the data of weight gain rate (WGR), specific growth rate (SGR) and relative food intake (RFI) with the gene expression (not yet published). Besides, our future research direction is to study the signaling pathway of apelin regulation on feeding and growth of common carp at both the gene and protein levels.’

Reviewer 2 Report
see attached document

Author Response
The manuscript entitled “The Regulatory Role of Apelin on the Appetite and Growth of Common Carp (Cyprinus Carpio L.)” presented the effects of apelin on the feed intake and growth through the measurement of gene expression. Before publishing Animal Journal, several issues need to be addressed.
Dear reviewer,
We appreciate the positive and constructive comments made by reviewers and editors on our manuscript (animals-952292) entitled “The Regulatory Role of Apelin on the Appetite and Growth of Common Carp (Cyprinus Carpio L.)” The comments are valuable and helpful for revising and improving our paper, as well as the important guiding significance to our researches. According to your suggestions, we have made correction which we hope to meet with approval. Based on your comments, we also attached a point-by-point letter to you. The main corrections in the manuscript and the responses to your comments are as following.
Yours sincerely,
Dr. Guoxing Nie (Corresponding author)
(On behalf of all co-authors)
The title: The Latin name of common carp was not correct. Cyprinus Carpio instead of Cyprimus Carpio.
Response:
Thanks for the reviewer’s kind advice. I'm very sorry for this spelling mistake and changed ‘Cyprimus Carpio’ to ‘Cyprinus Carpio’ in the title.
Line 13- 15: Summary: The authors only test one form of apelin (Pyr-apelin-13) in this study, the sentence should be rephased as “ Pyr-apelin-13 could induce significant changes in mRNA levels of appetite-related and growth-related genes, suggesting that apelin may regulate the appetite and growth of common carp by regulating the expression of these key genes”
Response:
Thanks for the reviewer’s kind advice. We have made corresponding changes according to your suggestion and rewrote this sentence. ‘Based on our investigations, in vitro and in vivo experiments showed that Pyr-apelin-13 could induce significant changes in mRNA levels of appetite-related and growth-related genes, suggesting that Pyr-apelin-13 may regulate the appetite and growth of common carp by regulating the expression of these key genes.’
Abstract: The aim of the study was not mentioned in this section. The name of genes needs to be in Italics. Please add the Latin name of common carp.
Response:
Thanks for the reviewer’s kind advice. We have made corresponding changes according to your suggestion and rewrote this section. Firstly, the purpose of the study was supplemented in the abstract. Secondly, make sure that all gene names are in Italics and the entire names were added. Then the Latin name of common carp (Cyprimnus Carpio L.) were added. As shown below:
‘Abstract: Apelin, a kind of active polypeptide, has many biological functions, such as promoting food intake, enhancing immunity and regulating energy balance. In mammal, studies have indicated that apelin involved in the regulating food intake. However, there are relatively few studies about the regulatory effect of apelin on fish feeding, and the specific mechanism is not clear. Therefore, the purpose of this study was to preliminarily investigate the regulatory effects of apelin on key genes of feeding and growth in common carp (Cyprinus Carpio L.) through in vitro and in vivo experiments. In the present study, after incubation with different concentration of Pyr-apelin-13 (0, 10, 100, 1000 nM) in hypothalamic fragments, the expressions of Neuropeptide Y (NPY) and Agouti related peptide (AgRP) mRNA were significantly up-regulated at 12 h and 3 h, respectively, and the down-regulation of Cocaine and amphetamine-related transcript (CART) mRNA expression was sharply observed at 1 h and 3 h. In vivo, after Pyr-apelin-13 oral administration (0, 1, 10, 100 pmol/g), orexin mRNA level in the hypothalamus of common carp was significantly promoted at 1 h, 6 h and 12 h, while CART/(Proopiomelanocortin) POMC mRNA level in the hypothalamus of common carp were significantly down-regulated. Following incubation with different concentration of Pyr-apelin-13 (0, 10, 100, 1000 nM) in primary hepatocytes, (Growth hormone receptor) GHR, (Insulin-like growth factor 2) IGF2, (Insulin like growth factor binding protein 2) IGFBP2 and (Insulin like growth factor binding protein 3) IGFBP3 mRNA level were significantly increased at 3 h. In vivo, the levels of (Insulin-like growth factor 1) IGF1, IGF2, (Insulin like growth factor binding protein 2) IGFBP2 and IGFBP3 mRNA were significantly increased after the oral administration of Pyr-apelin-13 in the hepatopancreas with time- and dose-dependent.’
These sentences need to rephase as the authors did not test all the form of apelin “These results indicated that apelin might regulate the feeding and the growth of common carp through mediating the expressions of appetite- and growth-related genes. Overall, apelin, which is an appetite-promoted factor, improves food intake and is involved in the growth of common carp”
Response:
Thanks for the reviewer’s kind advice. We have made corresponding changes according to your suggestion and rewrote this section. ‘These results support the hypothesis that Pyr-apelin-13 might regulate the feeding and the growth of common carp through mediating the expressions of appetite- and growth-related genes. Overall, apelin, which is an appetite-promoted factor, improves food intake and is involved in the growth of common carp.’
Line 40-41: Add the Latin name of the species
Response:
Thanks for the reviewer’s kind advice. We have made corresponding changes according to your suggestion and added the Latin name of the species.
In terms of amino acid composition, the 12 consecutive amino acids of goldfish (Carassius auratus) [4], common carp (Cyprinus Carpio L.) [5], and Ya-fish (Schizothorax prenanti) [6] at the C-terminal are exactly the same as that of vertebrates, which is also in the apelin-13 region.
Line 54: Pyr-apelin-13 should be use throughout the manuscripts
Response:
Thanks for the reviewer’s kind advice. We have made corresponding changes according to your suggestion and used the Pyr-apelin-13 in the results of our research and in the discussion that involving our results throughout the manuscripts.
Line 64-66: The genes name needs to be in Italics.
Response:
Thanks for the reviewer’s kind advice. ‘Orexin, agouti related peptide (AgRP), neuropeptide Y (NPY), Cocaine and amphetamine-related transcript (CART) and Proopiomelanoncortin (POMC) are important appetite regulators secreted from brain [12-14].’ In this sentence, Orexin, AgRP, NPY, CART and POMC are all proteins, so we don't use italics to represent them. When the gene expression of these proteins was detected in the manuscript, Italics were used according to your requirements.
Line 67-68: Add the references.
Response:
Thanks for the reviewer’s kind advice. We added the references in this sentence according to your suggestion. ‘Therefore, the results suggested that apelin regulated fish feed intake by regulating related endocrine factors [25, 26].’
Line 85- 86: How many common carp used in the experiment?
Response:
Thanks for the reviewer’s question. A total of 600 fish were purchased in this study, from which different specifications were selected for gavage experiment, hypothalamic incubation experiment, and hepatocyte incubation experiment, respectively.
Line 80: Commercial pellets from which company?
Response:
Thanks for the reviewer’s question. The commodity materials were purchased from Tongwei Co., Ltd. According to your suggestion, we have supplemented this content in the revised manuscript. ‘Fish were acclimated for 2 weeks in indoor tank (5×4×2 m3) and fed three times daily commercial pellets (Tong wei, China) until satiety before starting the trial [5].’
Line 85-86: This sentence needs to be rephased “To reduce the pain of the experimental fish, every effort was made and we anesthetized them with 100 mg/L MS-222 (Sigma, USA) before each treatment”
Response:
Thanks for the reviewer’s kind advice. We rephased this sentence. ‘To reduce the pain of the experimental fish, we anesthetized them with 100 mg/L MS-222 (Sigma, USA) before each treatment.’
Line 92: How many fish used in this experiment? How many tanks?
Response:
Thanks for the reviewer’s kind advice. We have made corresponding changes according to your suggestion and rewrote this section. ‘Ninety-six common carp with body weight of 97.55 ± 8.32 g were used for gavage administration of Pyr-apelin-13 test, and they were randomly divided evenly among 16 tanks.’
Line 93-94: The authors mentioned that fish were anesthetised by 100 mg/L MS-222 (Sigma, USA) and in this section the author mentioned that” experimental fish fasted 24 h were anesthetized with 0.05 % MS-222 (Aladdin Industrial Corporation, Shanghai, China)”. Is this correct?
Response:
Thanks for the reviewer’s question. Although the concentration of MS-222 we used in the experiment is uniform, we are very sorry that we used different statements in the same article. According to your suggestion, we have unified the expression method of MS-222 concentration (100 mg/L) in the manuscript.
Line 98: “proximate composition” instead of “nutrient levels”? Vitamin premix instead of “Citamin premix”. How many percent of Nitrogen-Free Extract (NFE) contain in the diet? It should be added in the table 1”
Response:
Ingredients |
Contents (%) |
Proximate composition |
Contents (%) |
Fish meal |
2.0 |
Moisture |
9.05 |
Soybean meal |
25.0 |
Crude lipid |
9.56 |
Rapeseed meal |
20.0 |
Crude protein |
34.93 |
Rice bran meal |
10.0 |
Ash |
9.00 |
Rice bran |
10.0 |
Nitrogen-free extract |
29.26 |
Fish oil |
2.0 |
Note: 1) Vitamin premix supplied the following vitamin (kg): VA 800000 IU, VB1 1500 mg, VB2 1250 mg, VC 2.5 g, VD3 160000 IU, VE 15 g, VB12 4 mg, VK3 325 mg, VB6 1100 mg, Creatine 5.5 g, Folic acid 70 mg, Biotin 125 mg, Niacin 4 g, Pantothenic acid 4.5 g. 2) Mineral premix supplied the following minerals (kg): P 105 g, Ca 330 g, Mg 45 g, Fe 15 g, I 50 mg, Se 9 mg, Cu 0.35 g, Zn 3 g, Mn 1.5 g, Co 11 mg. |
|
Soybean oil |
3.0 |
||
Sesame meal |
5.0 |
||
Flour |
10.0 |
||
Wheat bran |
10.0 |
||
Ca(H2PO4)2 |
2.0 |
||
Vitamin premix1 |
0.2 |
||
Mineral premix2 |
0.6 |
||
Choline chloride |
0.2 |
||
Total |
100.0 |
Thanks for the reviewer’s kind advice. We have made corresponding changes according to your suggestion and rewrote table 1. We changed ‘nutrient levels’ to ‘proximate composition’. We changed ‘Citamin premix’ to ‘Vitamin premix’. And Nitrogen-Free Extract (NFE) contain was added in the table 1.
Table 1 Formulation and proximate composition of the diet.
Line 100: How many fish? Please provide more details of experimental design.
Response:
Thanks for the reviewer’s question. Ninety-six fish were used in the experiment. And we added more details of experimental design. As shown below:
‘Experimental fish (n=96, 29.18 ± 2.31 g) were sacrificed with a severed head. The hypothalamus was quickly separated and placed into precooled Ca2+/Mg2+-free HBSS. In the ultra-clean workbench, the hypothalamus was rinsed with Ca2+/Mg2+-free HBSS three times to remove impurities, such as blood clots. The hypothalamus was divided into two pieces and then preincubated with containing 10 % FBS (Hyclone, USA) DMEM/F-12 medium (Gibco, USA) for 6 h at 28 °C and no CO2 [24]. After preincubation, the old medium was removed and the fragments of hypothalamus were rinsed twice with DMEM/F12 (without FBS), and then 500 μL DMEM/F-12 medium (Gibco, USA) was added. After 1 h, 500 μL medium (containing Pyr-apelin-13) was added into each well, with final concentrations of Pyr-apelin-13 (0, 10, 100, 1000 nM), respectively. After incubation for 1, 3, 6, 12 h, the hypothalamus fragments were analyzed by RNAiso Plus (TaKaRa, Japan), and quickly stored at −80 °C until analysis.’
Line 104: Add more details of serum product, which company?
Response:
Thanks for the reviewer’s question. FBS (Hyclone, USA) and DMEM/F-12 medium (Gibco, USA).
Line 110: “The” instead of “the”
Response:
Thanks for the reviewer’s question. ‘the’ was changed to ‘The’.
Line 122-126: Provide more details of qPCR. How many cycles? Which master mix, etc?
Response:
Thanks for the reviewer’s question. According to your suggestions, the relevant contents were supplemented as follows: ‘The PCR amplification was carried out using the LightCycler 480 II (Roche, Switzerland) with SYBR® Premix Ex TapTM II (Takara, Japan). All reactions were performed in triplicate. The procedure was as follows: 10 s at 95 °C for initial denaturation, followed by 40 cycles of 95 °C for 5 s, 60 °C for 20 s.’
Line 129: Details of SPSS (IBM?)
Response:
Thanks for the reviewer’s question. SPSS company information was supplemented in the revised manuscript. ‘And all data were analyzed using the software SPSS 20.0 (IBM, USA).’
Line 130: “range” instead of “rang”
Response:
Thanks for the reviewer’s kind advice. We have modified this part.
Line 135-137: Move those sentences to Method section.
Response:
Thanks for the reviewer’s kind advice. According to your suggestions, those sentences were removed.
Line 166: “appetite related genes” Why it need to be in Italics?
Response:
Thanks for the reviewer’s question. This is a formatting error caused by our negligence. These words should not be italicized.
Line 193: References needed.
Response:
Thanks. According to the suggestion of reviewer 1, we have deleted the first half of this paragraph.
Line 197: Pyr-apelin-13?
Response:
Thanks for the reviewer’s question. Apelin-13 and Pyr-apelin-13 are the two most commonly used forms of apelin in research. Here, the reagent used by the author [4, 17] is apelin-13.
Line 200: Genes name in Italics
Response:
Thanks for the reviewer’s kind advice. In this sentence, ‘Apelin, NPY, AgRP, CART and Orexin secreted by the brain are important appetite regulators.’ Apelin, NPY, AgRP, CART and Orexin are proteins secreted by the brain, so italics are not suitable here. When the gene expression of these proteins was detected in the manuscript, Italics were used according to your requirements.
Line 203- 213: This paragraph is not clear what the authors want to discuss. Please re-write it. Again authors did not test all the forms of apelin, please use pyr-apelin-13 instead of apelin.
Response:
Thanks for the reviewer’s kind advice. We have made corresponding changes according to your suggestion and rewrote this section. Meanwhile, We used Pyr-apelin-13 instead of apelin. (Details below)
‘Some studies have shown that orexin is an appetite-promoted polypeptide. And apelin can regulate the mRNA expression of orexin and then regulate fish feeding. For instance, in goldfish, Volkoff et al [25] studied the effect of apelin on orexin mRNA expression in hypothalamus, forebrain and afterbrain in vitro. It was found that the expression of orexin mRNA increased significantly after incubation with 100 nM apelin-13 for 1 h. In vivo, studies have shown that Pyr-apelin-13 can increase the feed intake by up-regulating orexin mRNA expression of the hypothalamus [26]. In the i.p. injection of cavefish, apelin significantly increased orexin mRNA level of the hypothalamus [19]. These results indicated that apelin can regulate fish feeding by up-regulating orexin mRNA expression. In our study, orexin mRNA expression of the hypothalamus increased significantly after Pyr-apelin-13 administration with dose-dependent in vivo. However, Pyr-apelin-13 in this study has no influence on the level of orexin mRNA in vitro. The difference of results may be related to species-specific, apelin concentration, apelin isoform (eg. apelin-12, apelin-13, apelin-36 or Pyr-apelin-13) or treatment way.’
Line: Gene name in Italics.
Response:
Thanks for the reviewer’s kind advice. In the revised manuscript, careful checks were made to ensure that the gene names were italicized and the protein names were not.
Line 224-229: Is there any other study in fish that the author can compare with?
Response:
Thanks for the reviewer’s question. According to the literature we consulted, we haven't found any reports about the regulation of POMC/CART by apelin in fish. Therefore, we are unable to provide any other studies on fish that can be compared for discussion temporarily. Next, we will continue to pay attention to relevant research reports.
Line 236: Reference needed.
Response:
Thanks for the reviewer’s kind advice. Regarding these sentences, reviewer 1 thinks that this sentence ‘Important information but no clear connections to this work.’ We have adopted his suggestion to delete these two sentences. Please refer to the revised draft for details.
Line 249: Latin name of Tilapia.
Response:
Thanks for the reviewer’s kind advice. When the word ‘Tilapia’ first appeared in the manuscript (Introduction), we added the Latin name of Tilapia, as you suggest. As shown below, ‘It was found that the food intake of tilapia (Oreochromis mossambicus) was significantly increased after intraperitoneal injection of NPY (0.6 g/g) for 10 h [21].’ Therefore, we use the common name of tilapia here.
Line 278: Discuss in more details the effect of pyr-apelin- 13 on IGFBP2 and IGFBP3.
Response:
Thanks for the reviewer’s kind advice. We have made corresponding changes according to your suggestion and rewrote this section (Details below). ‘Atlantic salmon (Salmo salar) parr were implanted intraperitoneally with cortisol implants (0, 10, and 40 μg/g body weight) and sampled after 3 or 14 days [34]. The results showed that hepatic IGFBP1 and IGFBP2 were stimulated by the high cortisol dose and indicated that cortisol modulates the growth of juvenile salmon via the regulation of hepatic IGFBPs [34]. In this study, Pyr-apelin-13 stimulated the expression of IGFBP2 and IGFBP3 both in vivo and in vitro. Therefore, combined with relevant research reports, it is speculated that Pyr-apelin-13 may regulate the growth of carp by regulating the expression of IGFBPs.’
Suggestion:
It would be nice if the authors can correlation the data of specific growth rate, feed intake with the gene expression.
Response:
Thanks for the reviewer’s kind advice. In our another study, common carp was fed with dietary supplementation of Pyr-apelin-13 for 8 weeks. We will correlation the data of weight gain rate (WGR), specific growth rate (SGR) and relative food intake (RFI) with the gene expression, according to your suggestion.
Please use “ feed intake” instead of “food intake” throughout the manuscripts
Response:
Thanks for the reviewer’s kind advice. We have made corresponding changes according to your suggestion and used ‘feed intake’ instead of ‘food intake’ throughout the manuscripts. Please refer to the revised draft for details.

Reviewer 3 Report
Dear authors,
increasing the appetite and thus feed intake in aquaculture species, is of interesting for the aquaculture industry. Your paper provides some data regarding the use of apelin as such a substance. However, I have severe doubts about the statistical evaluation of your data and strongly suggest to task an experienced statistician with the re-evaluation of your data. Since basically the whole results might change in case, the statistical evaluation reveals a different outcome, the overall results might change significantly.
In case the statistical re-evaluation does not change the results significantly, I still suggest to re-write large parts of your discussion. Providing data showing that gene expression of appetite and growth related genes are up-regulated does only give a hint of what might happen in case the substances are applied under pracitcal conditions. For pracitcal conditions apelin needs to be applied to huge amounts of fish in a time and labour sparing way. This is normally done via the feed. This would require apelin to be included into the diets as a feed additive, which brings its own set of open questions. Therefore you need to change your discussion accordingly and include these important points.
For the time being, I will not recommend to publish your paper in Animals and encourage you to work on your manuscript keeping the above comments in mind.
Kind regards
Author Response
Comments and Suggestions for Authors
Dear authors,
increasing the appetite and thus feed intake in aquaculture species, is of interesting for the aquaculture industry. Your paper provides some data regarding the use of apelin as such a substance. However, I have severe doubts about the statistical evaluation of your data and strongly suggest to task an experienced statistician with the re-evaluation of your data. Since basically the whole results might change in case, the statistical evaluation reveals a different outcome, the overall results might change significantly.
In case the statistical re-evaluation does not change the results significantly, I still suggest to re-write large parts of your discussion. Providing data showing that gene expression of appetite and growth related genes are up-regulated does only give a hint of what might happen in case the substances are applied under pracitcal conditions. For pracitcal conditions apelin needs to be applied to huge amounts of fish in a time and labour sparing way. This is normally done via the feed. This would require apelin to be included into the diets as a feed additive, which brings its own set of open questions. Therefore you need to change your discussion accordingly and include these important points.
For the time being, I will not recommend to publish your paper in Animals and encourage you to work on your manuscript keeping the above comments in mind.
Kind regards
Dear reviewer,
We appreciate the positive and constructive comments made by reviewers and editors on our manuscript (animals-952292) entitled “The Regulatory Role of Apelin on the Appetite and Growth of Common Carp (Cyprinus Carpio L.)” The comments are valuable and helpful for revising and improving our paper, as well as the important guiding significance to our researches. According to your comments, we have made a big change and correction which we hope to meet with approval. The main corrections in the manuscript and the responses to your comments are as following.
First of all, Thank you very much for your suggestions on statistical analysis of data. According to your suggestions, we used several statistical methods (such as Turkey, Duncan, LSD, ANOVA) to analyze the data, and the statistical re-evaluation does not change the results significantly. In the revised manuscript, in order to present the results more simply and clearly, we only showed the results of one-way ANOVA, that is, the difference between the treatment group and the control group at a single time point.
Secondly, according to the opinions and suggestions of the three reviewers, we have made substantial modifications and adjustments to the content of the manuscript, including Abstract, Introduction, Materials and methods, Result, Discussion and Conclusion. (Details were marked in red in the revised manuscript). We discussed and speculated the conclusions based on the experimental datas obtained and relevant research reports. This is a tentative exploration and further research is still needed to elucidate the regulation mechanism of apelin on feeding and growth of common carp. We hope the revised manuscript can meet with your approval.
In addition, in our another study, Pyr-apelin-13 was included into the diets as a feed additive. The common carp was fed with dietary supplementation of Pyr-apelin-13 for 8 weeks. We correlation the data of weight gain rate (WGR), specific growth rate (SGR) and relative food intake (RFI) with the gene expression. The results showed that Pyr-apelin-13 could promote the feed intake and growth of common carp because of the increase of weight gain rate (WGR), specific growth rate (SGR) and relative food intake (RFI). And the feeding and growth related genes of common carp are regulated by Pyr-apelin-13. Therefore, according to your suggestion, we changed the discussion accordingly and include these important points. We hope the revised manuscript can meet with your approval.
Thank you again for reviewing our manuscript.
Yours sincerely,
Dr. Guoxing Nie (Corresponding author)
(On behalf of all co-authors)
